# Deep Semiparametric Learning

**Mika Sarkin Jain & Jack Lindsey** *
Stanford University
Stanford, CA 94309
{mjain4, jacklindsey}@stanford.edu

## Abstract

We introduce a semiparametric approach to deep learning. Inspired by complementary learning systems theory in cognitive neuroscience, our approach combines elements of parametric and nonparametric learning by giving a neural network access to a differentiable nearest-neighbor algorithm. Analysis of our model suggests that it is able to leverage the respective advantages of nonparametric and parametric methods. Our model displays robustness to domain adaptation, rapid learning on limited training sets, and well-clustered embeddings while retaining the expressive power and generalization capabilities characteristic of parametric methods. We demonstrate that our model relies more heavily on nearest-neighbors information in early training but better approximates a purely parametric model as training progresses.

## 1 Introduction and Related Work

We introduce a semiparametric approach to deep learning. Our approach combines elements of parametric and nonparametric learning by giving a feedforward neural network access to a differentiable nearest neighbor computation.

Nonparametric learning algorithms like nearest neighbor-based methods[1] store data-label pairs, which are subsequently used as canonical examples of particular classes during classification. Parametric learning methods like deep neural networks learn parameters (i.e. weights) which define all stages of a model's computation. These methods are typically more powerful but can suffer from large training data requirements, sensitivity to changes in task domain, and poor geometric interpretability of embedding spaces. By providing nearest-neighbor results as a feature to a deep neural network, we match the performance of parametric approaches while addressing some of these shortcomings.

Our approach is inspired by the theory of complementary learning systems in cognitive neuroscience (McClelland et al., 1995; Kumaran et al., 2016), which holds that a combination of episodic (in the hippocampus) and statistical (in the neocortex) learning is important for human task solving. The hippocampus rapidly incorporates new observations to solve tasks in an example-based way, while the neocortex gradually learns abstract rules with greater power and generality.

Our model differs from prior work in two significant ways. The first is our approach to the problem of learning a useful distance metric. The work of Hoffer and Ailon (2015) uses a triplet network to learn this embedding map, which can be used later for, say, classification tasks. Hu et al. (2015) show the applicability of this kind of metric learning for transfer and domain adaptation. Notably, however, in these models a metric is learned prior to its application (using, say, k-nearest neighbors) for a target task such as classification. Our model may be trained end-to-end on the target task, incentivizing the network to learn a metric suitable to the relevant problem. In this respect we adhere more closely to the recent work of Zoran et al. (2017), which also achieves this end-to-end property via differentiable boundary trees. Second, our model takes the results of nearest-neighbors in embedding space as a *feature* rather than an output. In this respect our model contrasts with

---

*Both authors contributed equally to this work.

[1]For ease of discussion, we will refer to nearest-neighbor retrieval as nonparametric even if its distance metric is learned using a parameterized model.

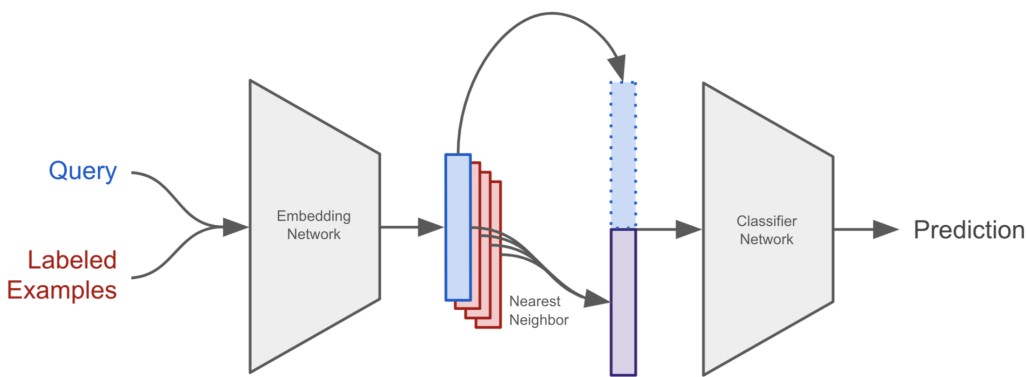

Figure 1: High-level schematic of our architecture. A neural network maps the query image, along with a batch of labeled examples, to an embedding space. A fully differentiable nearest-neighbor computation is performed on the batch, resulting in estimated class probabilities for the query image. A feedforward classifier takes as inputs these probabilites, along with the query image embedding.

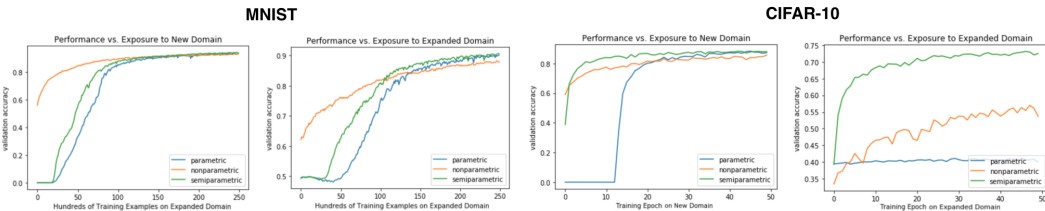

Figure 2: Model validation accuracy after exposure to $k$ training examples from new (unseen classes) or expanded (unseen and previously seen classes) domain.

recent work in memory-based parameter adaptation (Anonymous, 2018a), which neither considers nonparametric information at training time nor allows a network to learn how to use this information.

## 2 METHODS

We construct a hybrid parametric and non-parametric (i.e. "semiparametric") deep learning model designed to perform image classification. Our model consists of 1) a neural network that maps input to an embedding space 2) a fully differentiable nearest-neighbors-based classifier that operates in this embedding space and 3) a classifier network that operates on both the nearest-neighbors results and embedding space representation of an imput image. This model architecture is diagrammed in Figure 1 and our algorithm is specified in detail in Appendix A.

To make our model end-to-end trainable, we use a differentiable nearest-neighbors calculation that operates on data-label pairs of the current batch during training. Given the embedding $v_i$ of example $i$, we compute its squared distance $d_{ij} = (v_i - v_j)^2$ to the embedding of each example $j$, $j \neq i$, in the current batch. From these we compute weights $w_{ij} = d_{ij}^{-\tau}$ where $\tau$ is a hyperparameter that modulates the emphasis on tight clustering in embedding space (we have used $\tau = 2$). The calculation outputs class probabilities $P(c_i = a) = \sum_{j \neq i} w_j 1(c_j = a)$. These probabilities are concatenated with the embedding $v_i$ of the current example to form the input to the final classifier network.

We train our model on the MNIST, CIFAR-10, and CIFAR-100 datasets. Model details are found in Appendix B. We test against a parametric baseline, identical to the semiparametric model but without the nearest-neighbor step, and a nonparametric baseline which outputs the results of the nearest-neighbor step in the embedding space.

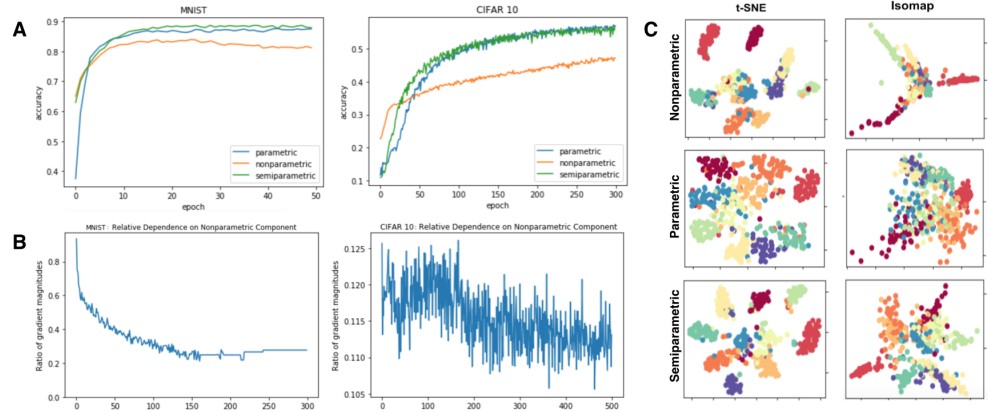

Figure 3: *A:* Validation accuracies on MNIST and CIFAR-10. *B:* The model's relative dependence on nearest-neighbors-based information, as measured by the ratio of gradient magnitudes with respect to the model's nearest-neighbor and parametric components. *C:* T-SNE and Isomap plots of the embedding space of each model after training, color-coded by class.

## 3 RESULTS

We first observe that the semiparametric model matches the learning trajectory and final performance of the parametric baseline model on MNIST, CIFAR-10, and CIFAR-100.

Next, we test each model on a domain adaptation problem. We train to convergence on a subset of MNIST containing half the available classes. Then we train it on k examples of unseen classes ("new domain"). varying k. We find (see Figure 2) that the nonparametric model gives good performance most quickly on the new domain and that the semiparametric model captures some of this advantage. The same results hold when we include all classes in the second ("expanded") domain, demonstrating that the semiparametric model can adapt without catastrophic interference. On a more difficult dataset, CIFAR-10, we performed a similar adaptation experiment but allowing multiple iterations over the training data in the second domain. We find that the semiparametric model adapts quickly. On CIFAR-10, the parametric baseline fails to learn on the expanded domain even after 500 epochs, indicating that it struggles to learn new categories without interfering with existing knowledge.

We experimented with training all models on 1000-image subsets on MNIST and CIFAR-10. As shown in Figure 3A, the nonparametric baseline learns quickly, but not asymptotically well; the parametric model has asymptotically better performance but initially requires more training time to learn. Our semiparametric model, on the other hand, learns both quickly and asymptotically well.

## 4 ANALYSIS

Our empirical results show that the semiparametric model mimics advantageous properties of the nonparametric model early in training but gradually converges toward the performance of the parametric model. We show directly that this phenomenon is due to initial reliance on the nonparametric component of the model which wanes over time. To measure this reliance, we compute the magnitude of the gradient of the model's output with respect to the output of the nearest-neighbors step, as a fraction of the magnitude of the gradient with respect to the embedding space. This metric serves as a first-order approximation to the model's relative dependence on nearest-neighbors retrieval. Figure 3B provides empirical evidence that this dependence wanes over time, consistent with the analogy to the psychological theory of complementary learning systems.

We analyzed the learned representations of each model, employing various low-dimensional embedding techniques (t-SNE and Isomap). (Figure 3C). The embeddings for each trained model map examples of the same class into local clusters. However, it also appears that the nonparametric model exhibits the tightest clustering, followed by the semiparametric model.

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

## 5  APENDIX A: ALGORITHM

---

**Algorithm 1** Deep Semiparametric Learning

---

1: $B$: batch size.
2: $(x_i, c_i)$: (example, label) pairs
3: $\tilde{c}$: one-hot encoding of class $c$
4: $\tau$: cluster separation hyperparameter (set to 2)
5: **for** each batch **do**
6:     **for** $i = 1, 2, \ldots, B$ **do**
7:         Map example $x_i$ into embedding $v_i$.
8:     **end for**
9:     **for** $i = 1, 2, \ldots, B$ **do**
10:         **for** $j = 1, 2, \ldots, B, j \neq i$ **do**
11:             $d_{ij} \leftarrow (v_i - v_j)^2$
12:             $w_{ij} \leftarrow (d_{ij})^{-\tau}$
13:         **end for**
14:         Estimate $\tilde{c}_i' = \sum_{j \neq i} w_{ij} c_j$
15:         Concatenate $v_i$ and $\tilde{c}_i'$, map to prediction $\hat{\tilde{c}}_i$
16:         Train predicted label $\hat{\tilde{c}}_i$ on actual label $c_i$
17:     **end for**
18: **end for**

---

# 6 APPENDIX B: METHOD DETAILS

On MNIST, we use a densely connected two-layer feedforward embedding network with hidden layer and embedding layer sizes of 32. Our classifier network consists of only one layer in this case. On the CIFAR-10 and CIFAR-100 examples, our embedding network consists two pairs of convolutional layers, with 32 and 64 channels respectively, and size 3 filters. We use size 2 max pooling and 25% dropout after each pair of convolutions. The embedding network ends with a dense layer into an embedding space of size 128. Our classifier network consists of two fully connected layers with hidden size 128. For all models, batches of size 100 were used on MNIST and 1000 on CIFAR. We employ ReLU nonlinearities for all non-terminal layers.

