# OpenReview forum: "Deep Semiparametric Learning"
_ICLR.cc/2018/Workshop — Reject_

### Official Review · AnonReviewer3 · 2018-03-08
**The insight is good, but it seems not practical**

**Rating:** 6
**Confidence:** 3

**Review:**

The manuscript proposed to concatenate the outputs from both parametric method (feed forward neural network) and nonparametric method (differentiable KNN) for making the final prediction, and all parameters are jointly trained.

The biggest advantage of this method is its ability to deal with newly generated classes or rare classes. Based on the results, new classes that have never been seen by the model can converge much faster, due to the nice property of KNN. The proposed method does not hurt the final performance when the training examples for new classes become more.

The downsides of the method is its computational cost. The manuscript seems not mentioning how to do inference. But I guess it needs to base its inference on some existing training examples due to the nonparametric part, which would increase the computational cost. Also, this method is doable for small number of unique classes. If the number of classes is much more than the batch size, then it is very likely that during training an example might not see any other examples that belong to the same class. It would be great if the manuscript could also show results for this scenario.

Questions:
- During inference, how to pick existing training examples for inference since a nonparametric method requires multiple training examples for inference?
- For nonparametric part, the output of an example is the weighted sum of the label of other examples. Do we need to normalize the weight? In the current manuscript, there is no normalizer.

---

### Official Review · AnonReviewer2 · 2018-03-10
**Interesting work but not good enough for acceptance**

**Rating:** 4
**Confidence:** 3

**Review:**

This paper proposes a semiparametric deep model which concatenates the nearest neighbors embedding (non-parametric) and instance embedding (parametric) for classification tasks. The experiments in domain adaption tried to argue that the proposed semiparametric model may take advantages of both parts, and converge quickly in the early stage and achieve satisfactory performance asymptotically.

Overall, the manuscript is well-written and easy to follow. The proposed approach is technically sound, but a bit trivial and not particularly novel as the idea of combining parameter and non-parametric methods is not quite new in literature. Also the experimental results are not convincing without comparison to any state of the art related works. The proposed semi-parameter approach is not always better, and the improvement is quite marginal.  The authors should provide more detailed descriptions about different experiment settings, the evaluation measure and more analysis of the inconsistence and negative results in experiments.

 (in Figure 2, such as why parametric model had no gain in the first 10 epochs. The authors claimed that the parametric baseline fails on CIFAR10 due to catastrophic forgetting, but you still trained them on expanded domain.

---

### Official Review · AnonReviewer1 · 2018-03-10
**Deep Semiparametric Learning**

**Rating:** 5
**Confidence:** 4

**Review:**

This paper proposes an interesting idea of combining direct memories (example-based learning) with statistal information about data (pattern based learning). This idea is borrowed from cognitive neuroscience. The model architecture has 3 main steps: 1) Embedding network over inputs in current batch , 2) Embedding is trained according to estimation of NNs in embedding space (differentiable) 3) Classifier that operates over a concatenation of the embedding of the input and the NN result.

Main idea and method are clearly presented, however, there are many missing details. Mainly, there is not information about the NN arquitectures and their main parameters. Furthermore, there is not sensibility analysis about these parameters.

It is not clear how is the final setup of the two baselines, a parametric and a non-parametric approach. Do the parametric approach still learns an initial embedding?, Do the non-parametric approach use plain Euclidean distance, (majority vote?, how to weight neighbors)?.

Results and analysis are interesting. In this sense, could be worth to show and discuss the ideas of the paper  in a ICLR workshop (although with an extended analysis).

---

### Decision · Program_Chairs · 2018-03-20
**ICLR 2018 Workshop Acceptance Decision**

**Decision:**

Reject

**Comment:**

Based on the reviews, this paper has not been accepted for presentation at the ICLR workshop. However, the conversation and updates can continue to appear here on OpenReview.